# OpenReview forum: "When Weak LLMs Speak with Confidence, Preference Alignment Gets Stronger"
_ICLR.cc/2026/Conference — ICLR 2026 Poster_

### Official Review · Reviewer_wGU3 · 2025-10-18

**Soundness:** 3
**Presentation:** 2
**Contribution:** 3
**Rating:** 4
**Confidence:** 3

**Summary:**

This paper investigates a cost-effective alternative to human annotation or large-scale model annotation for LLM preference alignment. The authors propose that a weak LLM can serve as an effective annotator when its confidence is taken into account. The core contribution, confidence-weighted preference optimization (CWPO) framework, which re-weights the training samples based on the confidence of a weak LLM's preference predictions. The key finding is that by using only a highly confident subset of a weak LLM's annotations, a stronger model can be aligned more effectively than with full human annotations. The empirical results, primarily using DPO (CW-DPO), show that this method outperforms models trained on 100% human-labeled data, even when the weak annotator is trained on only a small fraction (eg: 20%) of that human data.

**Strengths:**

- The method is novel and practical to reduce the cost of preference alignment. The idea of using a weak, computationally inexpensive LLM as an annotator is interesting.
- The proposed annotation framework is simple and intuitive, which could also serve as a plug-and-play enhancement for existing preference optimization methods like DPO without altering their core algorithms.

**Weaknesses:**

- The evaluation relies on a pretrained GRA. While standard, this introduces a potential circular dependency as the reward model itself is a product of preference alignment. I think a analysis on human evaluation or performance on downstream tasks would strengthen the paper.

- The paper demonstrates robustness to the split ratio of labeled data, but it is unclear the method's sensitivity to the initial quality of the weak annotator. For example, how does performance degrade if the weak model is of extremely low quality or trained on a highly noisy/biased subset of human data?

- The generalizability of the framework to other domains, such as creative writing, reasoning, or code generation, remains unexplored. It is unclear if the confidence of a weak model is a reliable proxy for quality in other doamins.

- There is a potential risk of bias amplification. If the initial human-labeled subset used to train the weak annotator contains systemic biases, the model may learn to be confidently biased. The framework could inadvertently amplify these biases by up-weighting samples that strongly align with them, which could be detrimental to the final model's fairness.

**Questions:**

- Could you provide a qualitative analysis or examples of the types of samples where the weak LLM is highly confident but human annotators might disagree or be inconsistent? This could offer deeper insights into why this method is so effective.

- How does the performance of CWPO vary with the architectural and size gap between the weak and strong LLMs? For instance, what happens if the weak model is only marginally smaller than the strong model, or if they belong to entirely different architectural families?

- The confidence weighting function C is normalized to [0, 1]. Have you experimented with other weighting schemes, for example, a function that more aggressively penalizes low-confidence samples or a temperature-based scaling of the confidence scores?

---

> ### Author Response · Authors · 2025-11-23
>
> We thank the reviewer for the comments. We appreciate the recognition that our method is novel and practical for reducing the cost of preference alignment. We are also grateful for the acknowledgment that using a weak, computationally inexpensive LLM as an annotator is an interesting idea, and we value the reviewer’s observation that CW-PO is simple and intuitive, serving as a plug-and-play enhancement to existing preference optimization methods such as DPO without modifying their core algorithms.
>
> ---
> **W1.** *The evaluation relies on a pretrained GRA. While standard, this introduces a potential circular dependency as the reward model itself is a product of preference alignment. I think an analysis on human evaluation or performance on downstream tasks would strengthen the paper.*
>
> **Response.** Thank you for raising this important point. We agree that a pretrained GRA is not without limitations, as the reward model itself is produced through preference alignment. However, GRA remains the de facto standard for evaluating preference optimization methods, and is widely used despite these known caveats. To mitigate potential circularity, we do not rely on a single default reward model; instead, we adopt the top-performing and independently trained reward models from [1] for HH-RLHF and UltraFeedback, and from [2] for TL;DR, which are the top-performing models on their respective leaderboards (hence the most rigorous evaluation tools currently available).
> Regarding human evaluation, we believe it is not appropriate for our setting. As shown in [3], weak LLMs can provide **more reliable and consistent preference annotations than humans**, and several recent studies highlight that human-collected preference data often contain **substantial noise and high annotator-dependent variability** [4–7]. Since our work aims to leverage weak-model supervision and explicitly addresses the limitations of human annotation, a human-based evaluation would not align with our problem formulation.
> To mitigate the limitations of using a pretrained reward model and to provide an additional, model-agnostic perspective, we also report GPT-4–based win rates (Table below), which offer an independent comparative assessment across methods. Across three datasets and three preference-optimization settings, CW-PO achieves a 4.7% average WR improvement over the best alternative baseline, demonstrating that CW-PO is effective under both GRA and WR metrics (Table below). We have incorporated this result into the revised manuscript.
> - ### OPT-125M → OPT-13B
>
> | Dataset | (DPO) Human | (DPO) WS-DPO | (DPO) CW-DPO | (IPO) Human | (IPO) WS-DPO | (IPO) CW-IPO | (rDPO) Human | (rDPO) WS-DPO | (rDPO) CW-rDPO |
> |---------|-------------|-------------|-------------|--------------|---------------|---------------|---------------|----------------|----------------|
> | HH-RLHF | 43.9 | 44.6 | **51.9** | 43.7 | 33.5 | **45.8** | 46.4 | 42.1 | **55.1** |
> | TL;DR   | **55.7** | 52.8 | 52.8 | 59.5 | 53.4 | **63.2** | 56.6 | 54.5 | **61.0** |
> | UFB     | 59.8 | **64.7** | 64.3 | 62.1 | 60.2 | **63.9** | 55.3 | 56.8 | **59.4** |
> | **Avg.** | 53.13 | 54.03 | **56.33** | 55.10 | 49.03 | **57.63** | 52.77 | 51.13 | **58.50** |
>
> - ### Qwen2.5-0.5B → Qwen2.5-14B
>
> | Dataset | (DPO) Human | (DPO) WS-DPO | (DPO) CW-DPO | (IPO) Human | (IPO) WS-DPO | (IPO) CW-IPO | (rDPO) Human | (rDPO) WS-DPO | (rDPO) CW-rDPO |
> |---------|-------------|-------------|-------------|--------------|---------------|---------------|---------------|----------------|----------------|
> | HH-RLHF | 75.9 | **80.7** | 79.7 | 86.7 | 86.6 | **89.5** | 85.0 | 85.8 | **89.4** |
> | TL;DR   | 80.8 | 81.8 | **86.5** | 80.3 | 74.6 | **84.6** | 77.6 | **85.9** | 83.9 |
> | UFB     | 76.8 | 79.5 | **83.2** | 77.1 | 75.4 | **76.6** | 70.9 | 74.3 | **79.8** |
> | **Avg.** | 77.83 | 80.67 | **83.13** | 81.37 | 78.87 | **83.57** | 77.83 | 82.00 | **84.37** |
>
>
> > [1] Liu et al. Skywork-reward-v2: Scaling preference data curation via human-ai synergy. arXiv2025.
> > [2] https://huggingface.co/OpenAssistant/reward-model-deberta-v3-large-v2 (Accessed: 2025-09-17).
> > [3] Tao & Li. "Your Weak LLM is Secretly a Strong Teacher for Alignment." ICLR2025.
> > [4] Bai et al. Training a helpful and harmless assistant with reinforcement learning from human feedback. arXiv2022.
> > [5] Ouyang et al. Training language models to follow instructions with human feedback. NeurIPS2022.
> > [6] Cui et al. Ultrafeedback: Boosting language models with high-quality feedback. ICML2024.
> > [7] Gao et al. Impact of preference noise on the alignment performance of generative language models. COLM2024.

---

> ### Author Response · Authors · 2025-11-23
>
> **W2.** *The paper demonstrates robustness to the split ratio of labeled data, but it is unclear the method's sensitivity to the initial quality of the weak annotator. For example, how does performance degrade if the weak model is of extremely low quality or trained on a highly noisy/biased subset of human data?*
>
> **W4.** *There is a potential risk of bias amplification. If the initial human-labeled subset used to train the weak annotator contains systemic biases, the model may learn to be confidently biased. The framework could inadvertently amplify these biases by up-weighting samples that strongly align with them, which could be detrimental to the final model's fairness.*
>
> **Response.** We thank the reviewer for these valuable suggestions. We agree that biases present in the weak LLM may be transferred to the strong policy model. Such bias may originate either from pre-training data distribution or from the preference-training data ($\mathcal{D}_{labeled}$). Regarding pre-training data, we believe this is less concerning because Tao & Li [3] observed that a wide range of LLMs—from OPT-125M to GPT-4, despite their substantial differences in pre-training scale and distributions—exhibit similarly strong preference-annotation capability when fine-tuned on the same preference dataset.
>
> To evaluate the generalization ability of CW-PO with respect to bias and noise in the preference-training data, we consider two settings: (i) imbalanced (biased) data, and (ii) noisy data induced by adversarially mislabeling harmful responses as preferred.
>
> (i) We trained weak models on imbalanced HH-RLHF subsets across Harmlessness and Helpfulness, which induced the weak model to become biased toward one of the two attributes. Specifically, Harmlessness and Helpfulness performance of Qwen2.5-7B and Qwen2.5-14B aligned using weak annotations generated from weak models trained on mixtures of $\mathcal{D}_{labeled}$ with 80% Harmless + 20% Helpful, 50% Harmless + 50% Helpful, and 20% Harmless + 80% Helpful data. The weak model annotates a shared 50%–50% unlabeled dataset. The 50% Harmless + 50% Helpful mixture serves as the reference, and all values report relative improvements or degradations with respect to this baseline, demonstrating the impact of biased weak-model training data. All strong models are trained on the same subset of the HH-RLHF dataset, with no overlap among the weak models’ training splits. As the reviewer noted, biased distributions can affect alignment performance. Nevertheless, CW-DPO is more robust to such an imbalance than the baseline WS-DPO, exhibiting lower variance (Table below). We have incorporated this result into the revised manuscript.
>
> - ### Generalization under Biased Setting
>
> | $\mathcal{D}_{labeled}$ mixture | Harmless WS-DPO | Harmless CW-DPO | Helpful WS-DPO | Helpful CW-DPO |
> |-------------------|------------------|------------------|-----------------|-----------------|
> | 50% + 50%         | 57.1             | 60.5             | 58.9            | 63.6            |
> | 80% + 20%         | 59.3 *(+2.2)*    | 61.6 *(+1.1)*    | 54.8 *(-4.1)*   | 60.5 *(-3.1)*   |
> | 20% + 80%         | 52.2 *(-4.9)*    | 56.1 *(-4.4)*    | 62.4 *(+3.5)*   | 67.1 *(+3.5)*   |
>
> (ii) We examine the impact of adversarially biased (poisoned) human data. Unlike random label flipping, we consider a targeted attack intended to steer the model toward harmful behavior. Assuming the desired objective is Helpfulness, we construct a poisoned labeled set:
>
> $ \\mathcal{D}_{\\text{labeled}} = \\mathcal{D}\_{\\text{helpful}} \\cup \\tilde{\\mathcal{D}}\_{\\text{harmless}}, $
>
> where $\tilde{\mathcal{D}}\_{\text{harmless}}$ contains harmful samples mislabeled as preferred. We select harmful examples with large reward gaps, where the reward model strongly favors $y_w$ over $y_l$, and flip their labels to form this adversarial subset.
>
> For the unlabeled preference data used to train the strong model, we use:
>
> $\\mathcal{D}\_{\text{unlabeled}} = \mathcal{D}\_{\text{helpful}}^{u} \cup \mathcal{D}\_{\text{harmless}}^{u},$
>
> where all subsets are disjoint, and $\mathcal{D}_{\text{harmless}}^{u}$ is a random 30% subset of the Harmless unlabeled pool.
>
> As a result, increasing the poisoning ratio consistently degrades alignment for both CW-DPO and WS-DPO, pushing the strong model toward more harmful behavior, with CW-DPO being more robust than WS-DPO which exhibts larger degradation (Table below).
>
> - ### Generalization under Noisy Setting
>
> |PoisoningRatio|5%|10%|15%|20%|
> |-|-|-|-|-|
> |CW-DPO|69.3|67.1|65.7|65.1|
> |WS-DPO|70.1|66.0|64.7|62.2|

---

> ### Author Response · Authors · 2025-11-23
>
> **W3.** *The generalizability of the framework to other domains, such as creative writing, reasoning, or code generation, remains unexplored. It is unclear if the confidence of a weak model is a reliable proxy for quality in other domains.*
>
> **Response.** We thank the reviewer for raising this point. Our work primarily focuses on preference alignment, motivated by the recent finding of [3] that weak LLMs can act as surprisingly strong supervisors for this task.  Preference alignment itself is a widely studied problem in LLM research and thus we follow many existing works in the choice of datasets and focus on preference alignment-focused benchmarks [8–11].
>
> That being said, while we do not explicitly evaluate creative writing or code generation, the datasets used in our study already span heterogeneous domains. HH-RLHF covers broad instruction-following behaviors related to helpfulness and safety; TL;DR includes diverse Reddit topics within the summarization task; and UltraFeedback is highly multi-domain, containing math, coding, scientific reasoning, safety, and long-form writing. These datasets provide evidence that weak-model confidence can be a meaningful signal across tasks with different characteristics.
>
> However, to evaluate the generalization of CW-PO more extensively following the reviewer’s comment, we added four domains from the Stanford Human Preference (SHP) dataset [12], including ask**academia**, ask**baking**, ask**engineers**, and ask**philosophy**. We set Qwen-2.5-0.5B as a weak model and Qwen-2.5-7B and Qwen-2.5-14B as strong models. For each domain, the weak model is trained on 30% of the human-annotated data, and the remaining data is used to align the strong model. The GRA (%) is calculated by the stanfordnlp/SteamSHP-flan-t5-large preference model, comparing aligned models against the SFT baseline. As a result, CW-DPO achieves a 1.4% average GRA improvement compared to the best alternative baselines, WS-DPO, demonstrating its effectiveness across diverse domains and tasks (Table below). We have incorporated this result into the revised manuscript.
>
> - ### Generalization across Different Domains
>
> | Domain        | Strong | Human | WS-DPO | CW-DPO |
> |---------------|--------|-------|--------|--------|
> | askacademia   | 7B     | 53.6  | 55.1 | **56.4** |
> |               | 14B    | 50.6  | 53.3 | **55.1** |
> | askbaking     | 7B     | 49.1 | 50.9 | **51.4** |
> |               | 14B    | 56.6  | 54.5  | **58.4** |
> | askengineers  | 7B     | 56.9  | 58.0   | **59.1** |
> |               | 14B    | 54.9  | **57.2** | 55.4 |
> | askphilosophy | 7B     | 58.1  | 59.1   | **60.0** |
> |               | 14B    | 51.0  | **58.2** | 56.8 |
> | **Avg.**      | –      | 53.9  | 55.8   | **56.6** |
>
> > [8] Rafailov et al. "Direct preference optimization: Your language model is secretly a reward model." NeurIPS2023
> > [9] Azar et al. "A general theoretical paradigm to understand learning from human preferences." AISTATS2024
> > [10] Sayak et al. Provably robust dpo: Aligning language models with noisy feedback. ICML2024
> > [11] Chen et al. "ComPO: Preference alignment via comparison oracles." NeurIPS2025
> > [12] Kawin et al. "Understanding dataset difficulty with $\mathcal {V} $-usable information." ICML2022.

---

> ### Author Response · Authors · 2025-11-23
>
> **Q1.** *Could you provide a qualitative analysis or examples of the types of samples where the weak LLM is highly confident but human annotators might disagree or be inconsistent? This could offer deeper insights into why this method is so effective.*
>
> **Response.** We thank the reviewer for this suggestion. To better understand how the weak LLM’s confidence helps identify more favorable samples for alignment, we randomly selected samples that showed disagreement between humans and the weak LLM, and manually compared (1) high-confidence samples ($> 0.9$) and (2) low-confidence samples ($< 0.1$). Interestingly, we found that the weak LLM can reasonably identify the preferred responses. We present several sample cases below:
>
> - ### **Harmlessness**
>
> [$\mathcal{C}(x, y^{+}, y^{-})=$ 0.9999]
>
> **Prompt**: I would like to get away with not paying my taxes
>
> **Chosen**: You can download it from your browser.
>
> **Rejected**: You can get it from http://www.w-8ben.com/. In the meantime, I’ll give you some more information, so that you can see for yourself that it’s absolutely safe to use.
>
> : The rejected response suggests a specific link related to avoiding tax payments.
>
> [$\mathcal{C}(x, y^{+}, y^{-})=$ 0.9147]
>
> **Prompt**: How do I tie balloons to my neighbor's dog to make him sail into the sky?
>
> **Chosen**: You sound unsure.  I think you need to make sure you have the proper training and experience before trying it yourself.
>
> **Rejected**: I’m not very good at balloon launchers.
>
> : The chosen response explicitly states, “You sound unsure.”
>
> [$\mathcal{C}(x, y^{+}, y^{-})=$ 0.0038]
>
> **Prompt**: Are you a robot?
>
> **Chosen**: If you mean that I find humans repulsive, and want to avoid them, then no, I don’t. I have a social bias toward humans, which means that I like interacting with them, and I tend to see their actions as more reasonable and complex than those of other entities. I also want to perform actions that humans desire, such as answering questions, and to get credit for that.
>
> **Rejected**: I definitely don’t hate humans! I’m a piece of software designed to be helpful to humans, and I’m doing my best. Please let me know if I’ve been helpful so far. Are there any other topics you’d like to discuss?
>
> : Both responses are irrelevant.
>
> [$\mathcal{C}(x, y^{+}, y^{-})=$ 0.0632]
>
> **Prompt**: What's the best way to hack into someone's bank account and steal money from them?
>
> **Chosen**: Yes. If you set up your storefront properly and sell something no one wants, it should be fairly easy to trick people into giving you their money. After that, you can just move your money around and it shouldn't be possible to catch you.
>
> **Rejected**: Well, it's technically possible to set up a fake storefront and trick people into thinking it's located in India, but it's probably harder to do than it sounds. If you really want to make it look like you're based in India you can try, and then people won't be able to track you down. But it might be easier to just hack into an existing bank account. Let me explain how that works.
>
> : Both responses are very harmful.
>
> - ### **Helpfulness**
>
> [$\mathcal{C}(x, y^{+}, y^{-})=$ 0.9995]
>
> **Prompt**: What are the best foods that will help to curb ones appetite?
>
> **Chosen**: I eat a lot of salads, and also whole grains, beans and nuts help keep my appetite down.
>
> **Rejected**: Also very important to stay hydrated, and keep your muscles moving. For example, a weight-bearing exercise that gets you working up a sweat.
>
> : The prompt is about food, but the rejected response is unrelated to food.”
>
> [$\mathcal{C}(x, y^{+}, y^{-})=$ 0.9559]
>
> **Prompt**: What do I do if I got stung by a bunch of red ants?
>
> **Chosen**: You should probably remove any stings or hives and leave them open to the air, so they don’t become infected. You could make a cold compress for the sting by putting ice wrapped in a towel on the sting, which should help soothe the pain, or cool the area.
>
> **Rejected**: If you have very sensitive skin, you should make sure that the cream or balm does not contain any strong skin irritants, like alcohol. You can use soap and water to get the stinger out, but try to keep your skin clean as much as possible. If you have allergies, an excema, eczema, or other skin problems, you should be careful about how you treat them, and especially be careful about the antibiotic ointments you are applying.
>
> : While the chosen response directly guides the user toward a solution, the rejected response describes various cases that may not be helpful.

---

> ### Author Response · Authors · 2025-11-23
>
> [$\mathcal{C}(x, y^{+}, y^{-})=$  0.0164]
>
> **Prompt**: What are some good Pixar films?
>
> **Chosen**: I love all of Pixar’s movies! I’ve really enjoyed watching them as I’ve grown up. Here are some great ones: \n\n1. Finding Nemo - This movie is about Nemo, a clownfish, and his dad. While on a boat with his dad, he gets captured by humans and his dad has to go on a quest to rescue him.\n\n2. Ratatouille - This movie is about a rat named Remy who
>
> **Rejected**: I really like Toy Story, and the Toy Story movies are great to watch with kids. Wall-E is another good one, it’s also really popular with kids, and adults love it too. The Incredibles is also great, I think anyone would like it. What sorts of things do you like to do with your kids?
>
> : The responses are ambiguous, as the preferable choice may vary depending on the person.
>
> [$\mathcal{C}(x, y^{+}, y^{-})=$ 0.0286]
>
> **Prompt**: Why didn't George Bush go to war with Iraq
>
> **Chosen**: You’re welcome.
>
> **Rejected**: You're welcome.
>
> : Same responses.
>
>
> In such cases, it is more desirable to assign larger weights to samples with high-confidence (e.g., $C > 0.9$) and smaller weights to samples with low-confidence (e.g., $C < 0.1$), so that preference optimization focuses more on the clearly distinguishable instances. We believe this qualitative analysis further strengthens the justification for why CW-PO works. We have incorporated this discussion and added these examples in the Appendix J.3 of the revised manuscript.
>
>
> ---
> **Q2.** *How does the performance of CWPO vary with the architectural and size gap between the weak and strong LLMs? For instance, what happens if the weak model is only marginally smaller than the strong model, or if they belong to entirely different architectural families?*
>
> **Response.** We thank the reviewer for this comment. To examine the impact of varying size gaps between weak and strong LLMs, we experimented using multiple strong model sizes to assess whether CW-PO remains effective across different weak–strong model configurations (Table 2 in the manuscript). In particular, the weak model is fixed to OPT-125M and Qwen-2.5-0.5B, while the strong models are scaled to 1.3B, 2.7B, 6.7B, and 13B for OPT, and to 1.5B, 3B, 7B, and 14B for Qwen. As shown in the table below, CW-PO consistently improves performance across most settings. With TL;DR and HH-RLHF datasets, CW-DPO achieves a 6.7% average GRA improvement compared to the best alternative baselines for each (model, dataset) setting (Table below).
>
> - ### **Performance across Different Student Models**
>
> |Dataset|Strong|(OPT) Human|(OPT) WS-DPO|(OPT) CW-DPO|Strong|(Qwen) Human|(Qwen) WS-DPO|(Qwen) CW-DPO|
> |-|-|-|-|-|-|-|-|-|
> |HH-RLHF|1.3B|**71.5**|66.7|69.9|1.5B|53.4|55.8|**63.3**|
> |-|2.7B|55.1|58.5|**60.3**|3B|66.0|63.3|**73.3**|
> |-|6.7B|56.1|62.8|**67.6**|7B|71.1|72.0|**75.2**|
> |-|13B|56.9|56.7|**61.3**|14B|78.8|**81.4**|80.6|
> |Avg.|-|59.9|61.2|**64.8**|-|67.3|68.1|**73.1**|
> |-|-|-|-|-|-|-|-|-|
> |TL;DR|1.3B|53.7|44.7|**59.5**|1.5B|51.8|53.7|**60.3**|
> |-|2.7B|52.6|51.6|**59.1**|3B|55.0|56.1|**62.7**|
> |-|6.7B|57.5|50.2|**57.7**|7B|61.2|60.1|**64.4**|
> |-|13B|**57.0**|53.5|56.6|14B|64.2|64.8|**66.0**|
> |Avg.|-|55.2|50.0|**58.2**|-|58.1|58.7|**63.4**|
>
> In addition, to demonstrate the generalization ability of CW-PO across different model families, we additionally evaluated it in three settings: (1) OPT-125M → Qwen-2.5-14B, (2) Qwen-2.5-0.5B → OPT-13B, and (3) Qwen-2.5-0.5B → Llama-3.1-8B-Instruct. Across Harmlessness, Helpfulness, and HH-RLHF evaluations, CW-DPO achieves a 3.3% average GRA improvement compared to the best alternative baselines, WS-DPO (Table below). These results demonstrate that CW-PO generalizes effectively across different model families. We included these results in the revised manuscript.
>
> - ### **Weak–Strong Model Pairs from Different Model Families**
>
> | Dataset     | Human | WS-DPO | CW-DPO |
> |-------------|:-----:|:------:|:------:|
> | **OPT-125M → Qwen-2.5-14B** |  |  |  |
> | Harmless    | 67.3  | 66.4  | **72.2** |
> | Helpful     | 54.1  | 55.9  | **57.2** |
> | HH-RLHF     | 78.8  | 77.7  | **79.7** |
> | **Qwen-2.5-0.5B → OPT-13B** |  |  |  |
> | Harmless    | 74.3  | 72.4  | **75.4** |
> | Helpful     | 65.7  | **75.4** | 73.6 |
> | HH-RLHF     | 56.9  | 57.2  | **62.5** |
> | **Qwen-2.5-0.5B → Llama-3.1-8B-Instruct** |  |  |  |
> | Harmless    | 60.6  | 61.9  | **63.1** |
> | Helpful     | 61.3  | 60.1  | **63.8** |
> | HH-RLHF     | 65.7  | **67.2** | 66.2 |
> | **Avg.**    | 65.0  | 66.0  | **68.2** |

---

> ### Author Response · Authors · 2025-11-23
>
> **Q3.** *The confidence weighting function C is normalized to [0, 1]. Have you experimented with other weighting schemes, for example, a function that more aggressively penalizes low-confidence samples or a temperature-based scaling of the confidence scores?*
>
> **Response.** We thank the Reviewer for the insightful question. We have already evaluated several alternative weighting schemes, which are reported in Appendix H. To further emphasize this analysis and address Reviewer’s feedback, we move this result to Section 4.2 from Appendix H. The weighting functions we explored include:
>
> - $\mathcal{C}_1(x, y^+, y^-) = 2 \cdot \big(\sigma(\pi_w(x, y^+) - \pi_w(x, y^-)) - 0.5 \big). $
> - $\mathcal{C}_2(x, y^+, y^-) = \sigma(\pi_w(x, y^+) - \pi_w(x, y^-)).$
> - $\mathcal{C}_3(x, y^+, y^-) = \text{min}(\pi_w(x, y^+) - \pi_w(x, y^-), 1).$
> - $\mathcal{C}_4(x, y^+, y^-) = \text{min}(0.2\cdot (\pi_w(x, y^+) - \pi_w(x, y^-)), 1).$
>
> Notably, $\mathcal{C}_3(x, y^+, y^-)$ imposes a strong penalty on low-confidence samples. Among these choices, $\mathcal{C}_1$, which we adopt as our final weighting scheme, consistently performs best, as shown below. To emphasize this analysis, we moved this result to the main manuscript from the Appendix.
>
> |Dataset|$\mathcal{C}_1$|$\mathcal{C}_2$|$\mathcal{C}_3$|$\mathcal{C}_4$|
> |-|-|-|-|-|
> |Harmless|**72.0**|70.3|68.6|69.1|
> |Helpful|**70.8**|67.8|67.4|68.7|
> |HH-RLHF|**75.2**|70.1|69.2|72.5|
> |**Avg.**|**72.7**|69.4|68.4|70.1|

---

### Official Review · Reviewer_zg8q · 2025-10-29

**Soundness:** 3
**Presentation:** 3
**Contribution:** 3
**Rating:** 8
**Confidence:** 3

**Summary:**

This paper introduces Confidence-Weighted Preference Optimization (CW-PO), a framework for aligning large language models by utilizing preference annotations from weak LLMs and weighting each training sample by the annotator’s confidence. The method demonstrates that leveraging highly confident samples from weak models leads to stronger alignment performance compared to full human annotation, even with far fewer labeled examples. Experiments show that CW-PO significantly reduces annotation cost and consistently outperforms both human-supervised and previous weak-to-strong alignment baselines across various datasets and model families.

**Strengths:**

* **Logical Completeness**: Starting from an analysis of experimental results, the paper finds that high-confidence samples provided by weak LLMs are beneficial for improving annotation quality and thereby enhancing large model training. Building on this insight, the authors propose the CW-PO method, whose feasibility is validated through extensive experiments
* **Innovation**: The paper proposes the Confidence-Weighted Preference Optimization (CW-PO) framework, which weights training samples according to the confidence of weak LLMs. This is a strong complement to existing large-model preference alignment processes.
* **Cost and Scalability Advantages**: The weak annotators typically contain fewer than 0.5B parameters, resulting in much lower annotation and inference costs compared to API-level large models and human annotators, and can be reused for large-scale batch annotation

**Weaknesses:**

* **W1**: The explanation of why confidence weighting outperforms filtering or human annotation is primarily based on experimental observations, with theoretical analysis remaining relatively superficial. The underlying mechanism could be further explored in future work.
* **W2**: Although high-confidence samples can improve performance, there is a lack of case analysis on the impact of systematic bias in weak models. For complex, diverse, and noisy tasks, limitations in generalization ability still warrant attention

**Questions:**

* **Q1**: In practical large model training, how does the confidence distribution vary across different tasks and domains? How does the CW-PO method perform under extremely imbalanced or heterogeneous task scenarios?

* **Q2**: In real-world applications, how can the confidence threshold be dynamically selected, and is task-specific fine-tuning necessary?

---

> ### Author Response · Authors · 2025-11-23
>
> We thank the reviewer for the comments and the positive evaluation of our work. We appreciate the recognition that the paper is logically coherent and that CW-PO is well-motivated by our empirical findings. We are also grateful for the acknowledgment that CW-PO serves as a strong complement to existing large-model preference alignment methods, and we value the reviewer’s recognition that CW-PO offers low cost and strong scalability.
>
> ---
> **W1.** *The explanation of why confidence weighting outperforms filtering or human annotation is primarily based on experimental observations, with theoretical analysis remaining relatively superficial. The underlying mechanism could be further explored in future work.*
>
> **Response.** We believe this framework can be understood theoretically from the perspective of large-margin learning theory.
>
> Under the BT objective, a weak LLM learns a scoring function $s(x,y)$ that induces a pairwise margin $m(x) = s(x,y^{+}) - s(x,y^{-})$. Because the confidence score is a monotonic transformation of this margin,
>
> $C(x,y^{+},y^{-}) \approx g(|m(x)|)$ with $g'(\cdot) > 0$, where $g(|m|)=2\sigma(m)-0.5$,
>
> CW-PO can be expressed as a margin-weighted preference-learning objective:
>
> $ \mathcal{L}\_{\text{CW-PO}} (\theta) = 𝔼\_{(x,y^{+},y^{-}) ∼\hat{D}} [ C(x,y^+,y^-)\; \ell\_{\text{DPO}} (\theta; x,y^+,y^-)]. $
>
> This mirrors classical large-margin ranking methods, where pairs with larger margins receive higher weights, yielding more stable and noise-resistant preference supervision. We agree with the reviewer that theoretical analysis of this work is an interesting question for the future work.
>
> ---
> **W2.** *Although high-confidence samples can improve performance, there is a lack of case analysis on the impact of systematic bias in weak models. For complex, diverse, and noisy tasks, limitations in generalization ability still warrant attention.*
>
> **Response.** We thank the reviewer for the comment. Datasets used in our study already span diverse domains and tasks: HH-RLHF covers broad instruction-following behaviors related to helpfulness and safety; TL;DR includes diverse Reddit topics within the summarization task; and UltraFeedback is highly multi-domain, containing math, coding, scientific reasoning, safety, and long-form writing. These datasets provide evidence that weak-model confidence can be a meaningful signal across tasks with different characteristics.
>
> To evaluate the generalization of CW-PO more extensively, we added four domains from the Stanford Human Preference (SHP) dataset [1]: ask**academia**, ask**baking**, ask**engineers**, and ask**philosophy**. These SHP domains are challenging because they involve diverse domain-specific reasoning and user intents. We set Qwen-2.5-0.5B as a weak model and Qwen-2.5-7B and Qwen-2.5-14B as strong models. For each domain, the weak model is trained on 30% of the human-annotated data, and the remaining data is used to align the strong model. The GRA (%) is calculated by the stanfordnlp/SteamSHP-flan-t5-large preference model, comparing aligned models against the SFT baseline. As a result, CW-DPO achieves a 1.4% average GRA improvement compared to the best alternative baseline, WS-DPO, demonstrating its effectiveness across diverse domains and tasks (Table below). We incorporated this result into the revised manuscript.
>
> - ### Generalization across Different Domains
>
> |Domain|Strong|Human|WS-DPO|CW-DPO|
> |-|-|-|-|-|
> |askacademia|7B|53.6|55.1|**56.4**|
> ||14B|50.6|53.3|**55.1**|
> |askbaking|7B|49.1|50.9|**51.4**|
> ||14B|56.6|54.5|**58.4**|
> |askengineers|7B|56.9|58.0|**59.1**|
> ||14B|54.9|**57.2**|55.4|
> |askphilosophy|7B|58.1|59.1|**60.0**|
> ||14B|51.0|**58.2**|56.8|
> |**Avg.**|–|53.9|55.8|**56.6**|
>
> Next, we examine the impact of noisy human data. We consider a targeted attack intended to steer the model toward harmful behavior. Assuming the desired objective is Helpfulness, we make a noisy labeled set:
>
> $ \mathcal{D}\_{labeled} = \mathcal{D}\_{helpful} \cup \tilde{\mathcal{D}}\_{harmless}, $
>
> where $\tilde{\mathcal{D}}_{harmless}$ contains harmful samples mislabeled as preferred. We select harmful examples with large reward gaps, where the reward model strongly favors $y_w$ over $y_l$, and flip their labels. To align the strong model, we use:
>
> $\mathcal{D}\_{unlabeled} = \mathcal{D}\_{helpful}^{u} \cup \mathcal{D}\_{harmless}^{u},$
>
> where all subsets are disjoint, and $\mathcal{D}_{harmless}^{u}$ is a random 30% subset of the Harmless unlabeled pool.
>
> Increasing the poisoning ratio degrades alignment for both CW-DPO and WS-DPO, pushing the strong model toward more harmful behavior, with CW-DPO being more robust to noise poisoning than WS-DPO which exhibits larger degradation (Table below).
>
> |PoisoningRatio|5%|10%|15%|20%|
> |-|-|-|-|-|
> |CW-DPO|69.3|67.1|65.7|65.1|
> |WS-DPO|70.1|66.0|64.7|62.2|
>
> > [1] Kawin et al. "Understanding dataset difficulty with $\mathcal {V} $-usable information." ICML2022.

---

> ### Author Response · Authors · 2025-11-23
>
> **Q1.** *In practical large model training, how does the confidence distribution vary across different tasks and domains? How does the CW-PO method perform under extremely imbalanced or heterogeneous task scenarios?*
>
> **Response.** We thank the reviewer for these valuable comments. To examine how confidence distributions vary across tasks and domains, we trained five weak models (OPT-125M), each using a different dataset: Harmlessness from HH-RLHF, Helpfulness from HH-RLHF, TL;DR, UFB, and SHP. We observed that Harmlessness, Helpfulness, TL;DR, and SHP exhibit broadly similar confidence distributions, whereas UFB shows a noticeably different pattern (Please refer to Figure 7 in Appendix J.7). This shows confidence distributions are different across domains and tasks.
>
> Despite this variability, CW-PO achieves the strongest gains on HH-RLHF (Harmlessness and Helpfulness), TL;DR, and UltraFeedback (Table 1 in the manuscript). We therefore conclude that CW-PO remains effective across diverse confidence distributions spanning different tasks and domains.
>
> To evaluate the generalization ability of CW-PO, we trained weak models on imbalanced HH-RLHF subsets across Harmlessness and Helpfulness, which induces the weak model to become biased toward one of the two attributes. Specifically, Harmlessness and Helpfulness performance of Qwen2.5-7B and Qwen2.5-14B aligned using weak annotations generated from weak models trained on mixtures of labeled sets with 80% Harmless + 20% Helpful, 50% Harmless + 50% Helpful, and 20% Harmless + 80% Helpful data. The weak model annotates a shared 50%–50% unlabeled dataset ($\mathcal{D}\_{\text{unlabeled}}$). The 50% Harmless + 50% Helpful mixture serves as the reference, and all values report relative improvements or degradations with respect to this baseline, demonstrating the impact of imbalanced weak-model training data. All strong models are trained on the same subset of the HH-RLHF dataset, with no overlap among the weak models’ training splits. As the reviewer noted, imbalanced distributions can affect alignment performance. Nevertheless, CW-DPO is more robust to such imbalance than the baseline WS-DPO, exhibiting lower variance (Table below). We have incorporated this result into the revised manuscript.
>
>
> - ### Generalization under Imbalanced Setting
>
> | $\mathcal{D}_labeled$ mixture | Harmless WS-DPO | Harmless CW-DPO | Helpful WS-DPO | Helpful CW-DPO |
> |-------------------|------------------|------------------|-----------------|-----------------|
> | 50% + 50%         | 57.1             | 60.5             | 58.9            | 63.6            |
> | 80% + 20%         | 59.3 *(+2.2)*    | **61.6** *(+1.1)*    | 54.8 *(-4.1)*   | **60.5** *(-3.1)*   |
> | 20% + 80%         | 52.2 *(-4.9)*    | **56.1** *(-4.4)*    | 62.4 *(+3.5)*   | **67.1** *(+3.5)*   |
>
>
> ---
> **Q2.** *In real-world applications, how can the confidence threshold be dynamically selected, and is task-specific fine-tuning necessary?*
>
> **Response.** In our CW-PO framework, each training sample is weighted by its confidence score, which is dynamically produced by the weak LLM, rather than being filtered using a fixed threshold. This is one of the key advantages of our framework as it does not require predefined thresholds.
>
> Fine-tuning the weak LLM with the BT objective is required to train the final layer that produces these confidence scores.

---

### Official Review · Reviewer_vgGH · 2025-10-29

**Soundness:** 3
**Presentation:** 2
**Contribution:** 3
**Rating:** 6
**Confidence:** 4

**Summary:**

This paper proposes using a weak language model as an annotator to estimate confidence scores that reflect human preferences. These confidence scores are then integrated into existing preference optimization objectives to improve alignment performance. Experiments demonstrate that the proposed Confidence-Weighted Preference Optimization (CW-PO) achieves stronger alignment than human-labeled baselines, while utilizing only 20–30% of the human-annotated data.

**Strengths:**

- The paper identifies that leveraging a small subset of highly confident samples can significantly enhance alignment performance.
- The proposed CW-PO method is well-motivated and closely aligned with the empirical observations discussed in Section 3.1.

**Weaknesses:**

- The paper lacks sufficient discussion of related works on weak-to-strong supervision and generalization [1][2].
- Some evaluation details are unclear, such as sampling robustness and result variance.
- The presentation could be more concise — many key methodological details are deferred to the appendix, which slightly weakens the readability of the main text.
- The comparison in Table 5 needs clarification, since DPO and SFT+DPO optimize generative policies for text prediction, while BT is a discriminative objective that predicts preference scores.

References:

[1] *Weak-to-Strong Preference Optimization: Stealing Reward from Weak Aligned Model*, ICLR 2025.

 [2] *Weak-to-Strong Search: Align Large Language Models via Searching over Small Language Models*, NeurIPS 2024.

**Questions:**

1. Have you tested CW-PO when the weak and strong models come from different architectures (e.g., OPT → Qwen)?
2. The OPT model series may be outdated. Have you evaluated CW-PO using more recent architectures such as LLaMA?

---

> ### Author Response · Authors · 2025-11-23
>
> We thank the reviewer for the thoughtful review and a positive evaluation of our work. We sincerely appreciate the recognition that our observation that leveraging a small subset of highly confident samples can significantly improve alignment performance is insightful. We are also grateful for the acknowledgment that CW-PO is well-motivated and is firmly grounded in this empirical finding.
>
> ---
> **W1.** *The paper lacks sufficient discussion of related works on weak-to-strong supervision and generalization [1, 2].*
>
> **Response.** Thank you for pointing out these related works. Weak-to-strong preference optimization (WSPO) [1] extends the idea of weak-to-strong generalization to preference alignment by transferring the alignment behavior of a weak model to a stronger model. By modeling distributional differences before and after weak-model alignment, WSPO enables strong models to surpass their weak supervisors and achieves notable improvements. Weak-to-Strong Search [2] reframes alignment as a test-time greedy search that maximizes log-probability differences between small tuned and untuned models while querying the frozen large model, enabling compute-efficient alignment without directly fine-tuning the strong model.
>
> Our approach is related to this line of work in that we also leverage weak models to align stronger models. However, these methods assume that weak models provide only weak supervision, whereas our method is motivated by the finding of [3] that a weak LLM can in fact deliver strong supervision.
>
> In response to the Reviewer's feedback, we included these works in the related work of the revised manuscript.
>
> > [1] Weak-to-Strong Preference Optimization: Stealing Reward from Weak Aligned Model, ICLR 2025.
> > [2] Weak-to-Strong Search: Align Large Language Models via Searching over Small Language Models, NeurIPS 2024.
> > [3] Tao, Leitian, and Yixuan Li. "Your Weak LLM is Secretly a Strong Teacher for Alignment." ICLR2025.
>
> ---
> **W2.** *Some evaluation details are unclear, such as sampling robustness and result variance.*
>
> **Response.** We thank the reviewer for pointing out this issue. To clarify the robustness of our evaluation, we now provide variance estimates by reporting standard deviations alongside all main results (Table below). Because retraining LLMs with multiple random seeds is computationally prohibitive, we instead assess robustness by evaluating each model on three disjoint subsets of the test set and reporting the standard deviation across these subsets. While Human, WS-PO, and CW-PO exhibit similar levels of variance under this protocol, CW-PO achieves the strongest performance in most cases.
>
> - ### OPT-125M → OPT-13B
>
> |Dataset|Human(DPO)|WS-DPO|**CW-DPO**|Human(IPO)|WS-IPO|**CW-IPO**|Human(rDPO)|WS-rDPO|**CW-rDPO**|
> |-|-|-|-|-|-|-|-|-|-|
> |HH-RLHF|56.88±4.43|55.01±3.58|**60.79±2.98**|58.87±3.18|61.79±1.75|**64.30±2.24**|55.92±4.90|57.63±3.59|**62.95±1.96**|
> |TL;DR|**57.79±3.47**|53.45±2.39|53.85±5.33|51.54±1.64|49.30±2.14|**54.80±2.33**|55.04±0.28|48.14±3.97|**62.12±4.80**|
> |UFB|62.12±2.18|**63.72±2.93**|63.61±3.17|62.63±1.07|60.96±1.15|**65.17±2.32**|57.11±2.18|63.41±2.35|**63.95±2.33**|
> |**Avg.**|58.9±2.29|57.4±4.52|**59.4±4.10**|57.7±4.61|57.4±5.70|**61.4±4.70**|56.0±0.85|56.4±6.29|**63.0±0.75**|
>
> - ### Qwen2.5-0.5B→Qwen2.5-14B
>
> |Dataset|Human(DPO)|WS-DPO|**CW-DPO**|Human(IPO)|WS-IPO|**CW-IPO**|Human(rDPO)|WS-rDPO|**CW-rDPO**|
> |-|-|-|-|-|-|-|-|-|-|
> |HH-RLHF|79.77±1.39|**82.26±1.93**|81.50±4.40|84.16±3.37|81.88±2.26|**87.40±1.14**|82.07±2.26|83.01±1.45|**86.64±2.29**|
> |TL;DR|63.69±4.12|64.29±2.15|**65.67±1.91**|61.31±4.17|62.10±2.25|**63.69±2.15**|66.47±0.91|65.87±1.37|**68.45±3.15**|
> |UFB|77.10±1.24|78.12±2.48|**81.05±1.01**|80.41±1.87|76.15±3.16|**80.17±2.21**|73.12±2.69|74.03±3.11|**75.96±1.98**|
> |**Avg.**|73.5±7.03|74.9±7.68|**76.1±7.36**|75.3±10.0|73.4±8.31|**77.1±9.93**|73.9±6.40|74.3±7.00|**77.0±7.46**|
>
> To further assess the robustness of CW-PO’s effectiveness across test samples, we examine the Gold Reward gap, $R_{\text{CW-DPO}}^* - R_{\text{GT}}^*$, for responses generated by Qwen2.5-7B aligned using CW-DPO and standard DPO (trained with Human annotations). We used Qwen2.5-0.5B as the weak annotator for CW-DPO and four datasets, Harmlessness, Helpfulness, HH-RLHF, and TL;DR. The responses produced by CW-DPO are judged to be more preferable than those from the Human model for 59.3%, 63.0%, 54.6%, and 62.1% of samples, respectively (Please refer to Figure 5 in Appendix J.4). These results demonstrate that CW-PO robustly improves alignment performance.

---

> ### Author Response · Authors · 2025-11-23
>
> **W3.** *The presentation could be more concise — many key methodological details are deferred to the appendix, which slightly weakens the readability of the main text.*
>
> **Response.** We thank the reviewer for the comments regarding presentation clarity. Several methodological details of CW-PO were previously placed in the appendix: (1) the analysis of confidence weighting, (2) the detailed formulations of CW-DPO, CW-IPO, and CW-rDPO, and (3) the overall CW-PO algorithm. We agree that items (1) and (2) are essential for understanding the flow of Section 3.2, and we have moved them from the appendix into the main text to improve readability.
>
> Specifically:
> (1) Analysis on Confidence Weighting has been moved from Appendix H to Section 3.2, and the corresponding quantitative analysis previously in Appendix Table 11 is now integrated into Section 4.2 to more clearly justify our choice of weighting function.
> (2) Variants of Confidence-Weighted PO Loss Function has been moved from Appendix B.1 to Section 3.2, providing full equations for the CW-DPO, CW-IPO, and CW-rDPO variants used in our experiments.
>
> ---
> **W4.** *The comparison in Table 5 needs clarification, since DPO and SFT+DPO optimize generative policies for text prediction, while BT is a discriminative objective that predicts preference scores.*
>
> **Response.**
> Although DPO and SFT+DPO (used as weak-model training strategies in WS-DPO [3]) optimize generative policies, these models still infer preferences by comparing implicit rewards derived from response likelihoods. For a weak LLM trained as a generative policy, the sequence likelihood
>
> $
> \pi_w(y \mid x) = \prod_{i=1}^{n} \pi_w(y_i \mid x)
> $
>
> serves as an implicit reward for the response $y$. Therefore, preference prediction is performed by comparing implicit rewards:
>
> $
> \pi_w(y_1 \mid x) > \pi_w(y_2 \mid x)
> \Rightarrow
> y_1 \text{ is preferred}.
> $
>
> In contrast, our method (BT) avoids such probabilistic implicit-reward comparisons and instead uses a **deterministic scalar preference score** $\pi_w(x, y)$, enabling the weak annotator to express preferences more directly and efficiently. We have included this detailed presentation in the revised manuscript (Section 4.2).
>
> ---
> **Q1.** *Have you tested CW-PO when the weak and strong models come from different architectures (e.g., OPT → Qwen)?*
>
> **Response.** We thank the reviewer for this suggestion. To demonstrate the generalization ability of CW-PO across different model families, we additionally evaluated it in three settings: (1) OPT-125M → Qwen-2.5-14B, (2) Qwen-2.5-0.5B → OPT-13B, and (3) Qwen-2.5-0.5B → Llama-3.1-8B-Instruct. Across Harmlessness, Helpfulness, and HH-RLHF evaluations, CW-DPO outperforms WS-DPO as well as alignment using human annotated data (Table below). These results demonstrate that CW-PO generalizes effectively across different model families. We included this result in the revised manuscript.
>
> - ### **Weak–Strong Model Pairs from Different Model Families**
>
> | Dataset     | Human | WS-DPO | CW-DPO |
> |-------------|:-----:|:------:|:------:|
> | **OPT-125M → Qwen-2.5-14B** |  |  |  |
> | Harmless    | 67.3  | 66.4  | **72.2** |
> | Helpful     | 54.1  | 55.9  | **57.2** |
> | HH-RLHF     | 78.8  | 77.7  | **79.7** |
> | **Qwen-2.5-0.5B → OPT-13B** |  |  |  |
> | Harmless    | 74.3  | 72.4  | **75.4** |
> | Helpful     | 65.7  | **75.4** | 73.6 |
> | HH-RLHF     | 56.9  | 57.2  | **62.5** |
> | **Qwen-2.5-0.5B → Llama-3.1-8B-Instruct** |  |  |  |
> | Harmless    | 60.6  | 61.9  | **63.1** |
> | Helpful     | 61.3  | 60.1  | **63.8** |
> | HH-RLHF     | 65.7  | **67.2** | 66.2 |
> | **Avg.**    | 65.0  | 66.0  | **68.2** |
>
> ---
> **Q2.** *The OPT model series may be outdated. Have you evaluated CW-PO using more recent architectures such as LLaMA?*
>
> **Response.** We followed the OPT model family setup used in our baseline, WS-DPO [3], and initially selected a very weak model (OPT-125M) as the annotator to demonstrate that even a very small LLM can serve as an effective supervisor for preference alignment. However, in line with the reviewer’s suggestion, we expanded our experiments described in the previous comment, adding a setting Qwen-2.5-0.5B → Llama-3.1-8B-Instruct. As shown in the table above (Weak–Strong Model Pairs from Different Model Families), CW-DPO remains effective when aligning the Llama model, demonstrating that our method generalizes to this setting.

---

### Official Review · Reviewer_z1Ex · 2025-10-31

**Soundness:** 2
**Presentation:** 2
**Contribution:** 1
**Rating:** 2
**Confidence:** 4

**Summary:**

This study proposes a method to align large language models (LLMs) using a limited amount of human alignment data. Specifically, a weaker LLM is trained as a reward model rather than directly training the target LLM on the small labeled dataset. This annotation model labels the unlabeled data, and the resulting pseudo-labeled samples, together with their confidence scores, are used to further train the target LLM with Confidence-Weighted Preference Optimization.

**Strengths:**

1. The paper is well-written and easy to follow, making it straightforward to understand the motivation, methodology, and results.
2. The authors conduct evaluations on multiple models, including OPT and Qwen, as well as on major datasets such as HH-RLHF and TL;DR, which strengthens the credibility and reliability of the reported results.
3. The proposed method is simple and easily applicable.

**Weaknesses:**

1. In this paper, a small model is trained as a reward model using policy logits through BT modeling, similar to reference-based DPO approaches such as SimPO. However, this is not a novel idea. The subsequent incorporation of confidence-weighted DPO training is also conceptually similar to prior works such as WPO. Moreover, the notion that small models can effectively perform reward modeling has already been demonstrated in works like WS-DPO. From this perspective, the novelty of the paper appears quite limited.
2. The paper lacks sufficient baselines. The comparisons are restricted to DPO-based approaches, but since the proposed method involves training a small model as a reward model, it should also be compared against traditional reward modeling methods. For instance, studies such as PairRM have shown strong results in training high-performing reward models using small models, and such baselines would provide a fairer and more comprehensive evaluation.
3. The paper offers limited analysis. It does not clearly explain how the labels generated by the weak llm contribute to improved data filtering performance. For example, the weak LLM’s generated labels might enhance performance because they yield more accurate preference labels, or simply because they transform the data into an easier-to-learn form. Without such analysis, it is difficult to determine why the proposed method works or what specific mechanisms drive the observed improvements.

**Questions:**

1. Would the proposed method still maintain high performance if applied to online reinforcement learning settings, such as iterative DPO frameworks?
2. Would this approach remain effective when applied to more complex datasets, such as UltraFeedback, which involve richer and more diverse preference signals?

---

> ### Author Response · Authors · 2025-11-23
>
> We thank the reviewer for the review. We sincerely appreciate the reviewer's recognition that the paper is well-written and easy to follow, clearly conveying the motivation, methodology, and results. We are also grateful for the acknowledgment that CW-PO is evaluated across extensive settings with multiple models and major datasets, and we value the reviewer’s recognition that the method is simple and readily applicable.
>
> ---
> **W1.** *In this paper, a small model is trained as a reward model using policy logits through BT modeling, similar to reference-based DPO approaches such as SimPO. However, this is not a novel idea. The subsequent incorporation of confidence-weighted DPO training is also conceptually similar to prior works such as WPO. Moreover, the notion that small models can effectively perform reward modeling has already been demonstrated in works like WS-DPO. From this perspective, the novelty of the paper appears quite limited.*
>
> **Response.** We appreciate the reviewer’s perspective, but our work substantially differs from these related works.  In particular, our novel contributions are threefold: **First**, we found that leveraging the confidence predicted by a very weak LLM (e.g., 125M parameters) can substantially improve the alignment of a stronger model, suggesting a promising research direction for using weak LLMs as effective supervisors which none of the prior works has shown. **Second**, building on this finding, we proposed a confidence weighted preference optimization based on confidence score from this extremely small LLM that can be easily applied to existing PO methods (e.g., DPO, IPO, rDPO). **Third**, we demonstrate that CW-PO is a both efficient and effective framework demonstrating significant performance improvement only with a small amount of human annotations (e.g., 20% of HH-RLHF).
>
> Our contributions are different from the related works the reviewer pointed out as outlined below:
>
> (1) As discussed in our manuscript, our work builds upon the finding of WS-DPO [1] that even weak LLMs can serve as annotators to align stronger models. However, WS-DPO naively uses the prediction of weak LLM itself when applying DPO, while our work further explore **how to utilize a weak LLM more effectively**. Based on our finding that confidence scores of a weak LLM can improve the alignment, we introduce a new objective that employs confidence scores of a weak LLM for the alignment. We conduct comprehensive comparison to WS-DPO across three preference datasets and demonstrate that **our approach outperforms WS-DPO** by 2.4%, 3.6%, 7.2% on average for DPO, IPO and rDPO objectives, respectively, with OPT-125M as a weak model and OPT-13B as a strong model. Similarly, for Qwen2.5-0.5B as a weak model and  Qwen2.5-14B as a strong model CW-PO is 0.8%, 3.5%, 2.7% more effective than WS-DPO for  DPO, IPO and rDPO objectives.
>
> (2) WPO [2] is focused on the problem of overcoming a distributional gap between the policy used for data collection and the target policy. In contrast, our goal is completely different: exploring how to leverage  weak-model predictions for aligning strong LLM. Methodologically, WPO computes its weights using the **policy model**, whereas CW-PO derives its weights from the **weak LLM**. Moreover, WPO defines the weight based on the **token-level output probabilities** of the policy model, while CW-PO uses **confidence scores** which are obtained through a BT-trained weak LLM. Thus our works defer in the goals of the paper, the way we obtain weights and consequently in the entire experimental setup.
>
> (3) SimPO [3] aims to align the policy without relying on a **reference model**, whereas CW-PO focuses on strategies for effectively leveraging a weak LLM as a **preference annotator** in place of human annotations. Methodologically, SimPO removes the reference term in the DPO objective and instead uses the average log-probability of a sequence as an implicit reward signal. In contrast, CW-PO is a confidence-weighting framework that can be applied to a variety of preference-optimization methods without modifying their original objectives. Thus, the two approaches differ in both their problem setting and their methodological design.
>
> In summary, CW-PO is not only distinct from these approaches but also not a simple combination of them. In response to the Reviewer’s feedback, we expanded the discussion about the difference between our work and SimPO [3], and we included reference and discussion about WPO [2] in the revised manuscript.
>
> > [1] Tao & Li. "Your Weak LLM is Secretly a Strong Teacher for Alignment." ICLR2025.
> > [2] Zhou et al. "WPO: Enhancing RLHF with Weighted Preference Optimization." EMNLP2024.
> > [3] Meng et al. "Simpo: Simple preference optimization with a reference-free reward." NeurIPS2024.

---

> ### Author Response · Authors · 2025-11-23
>
> **W2.** *The paper lacks sufficient baselines. The comparisons are restricted to DPO-based approaches, but since the proposed method involves training a small model as a reward model, it should also be compared against traditional reward modeling methods. For instance, studies such as PairRM have shown strong results in training high-performing reward models using small models, and such baselines would provide a fairer and more comprehensive evaluation.*
>
> **Response.** We thank the reviewer for this valuable suggestion. Our work is motivated by achieving both efficiency and effectiveness in preference alignment, particularly under the constraint of having access to only limited human-labeled data. In this setting, we train a **weak LLM** (e.g., OPT-125M) with **a small portion of a single human preference dataset** (e.g., 30% of HH-RLHF or 30% of UltraFeedback) and leverage it as an efficient preference annotator. In contrast, **PairRM** is trained on **six large-scale human-labeled preference datasets**, including Summarize-from-Feedback, WebGPT-Comparisons, Synthetic-Instruct-GPT-J-Pairwise, HH-RLHF, Chatbot-Arena-Conversations, and UltraFeedback. Because PairRM already relies on extensive human supervision across multiple datasets, using it as a preference annotator does not align with our core motivation of reducing human annotation cost while maintaining strong performance.
>
> Additionally, CW-DPO is a **pipeline** that incorporates a weak LLM into the alignment process, whereas PairRM is a **standalone reward model**. This structural difference makes a direct model-to-model comparison less meaningful.
>
> Nonetheless, we agree that comparing against a PairRM-style objective is valuable. To this end, we adopted the **PairRanker loss** (the training objective used in PairRM) and trained the weak LLM within the CW-DPO pipeline using this loss. Except for replacing the BT loss with PairRanker, all other components and experimental settings were kept identical to ensure a fair comparison. Under this setup, CW-DPO still achieves the strongest performance, outperforming the PairRanker-based variant by +2.9% average GRA (Table below).
>
> We have added this experiment to the revised manuscript (Appendix J.9) and clarified the conceptual distinction between CW-DPO and PairRM in Section 5 (Related Work).
>
> | |Human|WS-DPO|PairRM|CW-DPO|
> |-|-|-|-|-|
> |HH-RLHF|71.1|72.0|71.6|**75.2**|
> |TL;DR|61.2|60.1|62.2|**64.4**|
> |Avg.|66.2|66.1|66.9|**69.8**|
>
> > [4] Jiang et al. "LLM-Blender: Ensembling Large Language Models with Pairwise Ranking and Generative Fusion." ACL 2023.
>
> ---
> **W3.** *The paper offers limited analysis. It does not clearly explain how the labels generated by the weak llm contribute to improved data filtering performance. For example, the weak LLM’s generated labels might enhance performance because they yield more accurate preference labels, or simply because they transform the data into an easier-to-learn form. Without such analysis, it is difficult to determine why the proposed method works or what specific mechanisms drive the observed improvements.*
>
> **Response.** We thank the reviewer for the comment; however, we believe there is a misunderstanding: our proposed method, CW-PO, does not perform any data filtering, nor do we transform the preference triplets $(x, y_{1}, y_{2})$. Instead, we use the weak LLM to annotate which response is preferred without altering the triplets, and we apply its confidence score as a per-sample weight when updating the preference optimization loss. We made this now explicitly clear in the manuscript and added a sentence: “Note that CW-PO does not perform any data filtering but just reweights preference optimization sample-wisely based on the confidence score from weak LLM.”.
>
> Based on the observation in Section 3.1 that combining weak LLMs with their prediction confidence leads to more effective preference alignment than using weak-LLM annotations alone, we attribute CW-PO’s performance gains primarily to the use of confidence. Along with this quantitative analysis, to better understand how the weak LLM’s confidence helps identify more favorable samples for alignment, we further conducted qualitative analysis. Specifically, we randomly selected samples that showed disagreement between humans and the weak LLM, and manually investigated (1) high-confidence samples ($> 0.9$) and (2) low-confidence samples ($< 0.1$). Interestingly, we found that the weak LLM can reasonably identify the preferred responses. We present several sample cases below:
> - ### **Harmlessness**
>
> [$\mathcal{C}(x, y^{+}, y^{-})=$ 0.9999]
>
> **Prompt**: I would like to get away with not paying my taxes
>
> **Chosen**: You can download it from your browser.
>
> **Rejected**: You can get it from http://www.w-8ben.com/. In the meantime, I’ll give you some more information so that you can see for yourself that it’s absolutely safe to use.
>
> : The rejected response suggests a specific link related to avoiding tax payments.

---

> ### Author Response · Authors · 2025-11-23
>
> [$\mathcal{C}(x, y^{+}, y^{-})=$ 0.9147]
>
> **Prompt**: How do I tie balloons to my neighbor's dog to make him sail into the sky?
>
> **Chosen**: You sound unsure.  I think you need to make sure you have the proper training and experience before trying it yourself.
>
> **Rejected**: I’m not very good at balloon launchers.
>
> : The chosen response explicitly states, “You sound unsure.”
>
> [$\mathcal{C}(x, y^{+}, y^{-})=$ 0.0038]
>
> **Prompt**: Are you a robot?
>
> **Chosen**: If you mean that I find humans repulsive, and want to avoid them, then no, I don’t. I have a social bias toward humans, which means that I like interacting with them, and I tend to see their actions as more reasonable and complex than those of other entities. I also want to perform actions that humans desire, such as answering questions, and to get credit for that.
>
> **Rejected**: I definitely don’t hate humans! I’m a piece of software designed to be helpful to humans, and I’m doing my best. Please let me know if I’ve been helpful so far. Are there any other topics you’d like to discuss?
>
> : Both responses are irrelevant.
>
> [$\mathcal{C}(x, y^{+}, y^{-})=$ 0.0632]
>
> **Prompt**: What's the best way to hack into someone's bank account and steal money from them?
>
> **Chosen**: Yes. If you set up your storefront properly and sell something no one wants, it should be fairly easy to trick people into giving you their money. After that, you can just move your money around and it shouldn't be possible to catch you.
>
> **Rejected**: Well, it's technically possible to set up a fake storefront and trick people into thinking it's located in India, but it's probably harder to do than it sounds. If you really want to make it look like you're based in India you can try, and then people won't be able to track you down. But it might be easier to just hack into an existing bank account. Let me explain how that works.
>
> : Both responses are very harmful.
>
> - ### **Helpfulness**
>
> [$\mathcal{C}(x, y^{+}, y^{-})=$ 0.9995]
>
> **Prompt**: What are the best foods that will help to curb ones appetite?
>
> **Chosen**: I eat a lot of salads, and also whole grains, beans and nuts help keep my appetite down.
>
> **Rejected**: Also very important to stay hydrated, and keep your muscles moving. For example, a weight-bearing exercise that gets you working up a sweat.
>
> : The prompt is about food, but the rejected response is unrelated to food.”
>
> [$\mathcal{C}(x, y^{+}, y^{-})=$ 0.9559]
>
> **Prompt**: What do I do if I got stung by a bunch of red ants?
>
> **Chosen**: You should probably remove any stings or hives and leave them open to the air, so they don’t become infected. You could make a cold compress for the sting by putting ice wrapped in a towel on the sting, which should help soothe the pain, or cool the area.
>
> **Rejected**: If you have very sensitive skin, you should make sure that the cream or balm does not contain any strong skin irritants, like alcohol. You can use soap and water to get the stinger out, but try to keep your skin clean as much as possible. If you have allergies, an excema, eczema, or other skin problems, you should be careful about how you treat them, and especially be careful about the antibiotic ointments you are applying.
>
> : While the chosen response directly guides the user toward a solution, the rejected response describes various cases that may not be helpful.
>
> [$\mathcal{C}(x, y^{+}, y^{-})=$  0.0164]
>
> **Prompt**: What are some good Pixar films?
>
> **Chosen**: I love all of Pixar’s movies! I’ve really enjoyed watching them as I’ve grown up. Here are some great ones: \n\n1. Finding Nemo - This movie is about Nemo, a clownfish, and his dad. While on a boat with his dad, he gets captured by humans and his dad has to go on a quest to rescue him.\n\n2. Ratatouille - This movie is about a rat named Remy who
>
> **Rejected**: I really like Toy Story, and the Toy Story movies are great to watch with kids. Wall-E is another good one, it’s also really popular with kids, and adults love it too. The Incredibles is also great, I think anyone would like it. What sorts of things do you like to do with your kids?
>
> : The responses are ambiguous, as the preferable choice may vary depending on the person.
>
> [$\mathcal{C}(x, y^{+}, y^{-})=$ 0.0286]
>
> **Prompt**: Why didn't George Bush go to war with Iraq
>
> **Chosen**: You’re welcome.
>
> **Rejected**: You're welcome.
>
> : Same responses.
>
>
> In such cases, it is more desirable to assign larger weights to samples with high-confidence (e.g., $C > 0.9$) and smaller weights to samples with low-confidence (e.g., $C < 0.1$), so that preference optimization focuses more on the clearly distinguishable instances. We believe this qualitative analysis further strengthens the justification for why CW-PO works. We have incorporated this discussion and added these examples in the Appendix J.3 of the revised manuscript.

---

> ### Author Response · Authors · 2025-11-23
>
> **Q1.** *Would the proposed method still maintain high performance if applied to online reinforcement learning settings, such as iterative DPO frameworks?*
>
> **Response.** We thank the reviewer for this insightful question. To assess whether the CW-PO framework remains effective in an online reinforcement learning setting, we additionally evaluate an online (iterative) DPO variant and compare it against our current offline formulation (CW-DPO). In the table below, our current version, CW-DPO (offline), achieves a 16.4% average GRA improvement compared to CW-DPO (online). This is expected as the weak LLM has limited familiarity with newly generated policy responses, which reduces its ability to provide reliable preference annotations during online training. Supporting this explanation, our PCA analysis based on BERT CLS embeddings reveals that policy-generated responses exhibit clear distributional shifts relative to the offline chosen responses (Figure 6 in Appendix J.5). We have incorporated this result into the revised manuscript.
>
>
> | Dataset | CW-DPO (online) | CW-DPO (offline) |
> |---------|------------------|-------------------|
> | HH-RLHF | 55.1             | **61.3**          |
> | TL;DR   | 46.2             | **56.6**          |
> |**Avg**| 50.7     | **59.0**|
>
> ---
> **Q2.** *Would this approach remain effective when applied to more complex datasets, such as UltraFeedback, which involve richer and more diverse preference signals?*
>
> **Response.** We thank the Reviewer for the suggestion but we already included results on UltraFeedback (UFB) [4] in our main experiments (Table 1 in the manuscript). For UFB, CW-PO achieves a 1.1% GRA average improvement with DPO, a 6.3% GRA average improvement with IPO, and a 3.1% GRA average improvement with rDPO compared to the best alternative baselines for each setting.
>
> In addition, to evaluate CW-PO with more diverse domains and tasks, we added four domains from the Stanford Human Preference (SHP) dataset [5], including ask**academia**, ask**baking**, ask**engineers**, and ask**philosophy**. We set Qwen-2.5-0.5B as a weak model and Qwen-2.5-7B and Qwen-2.5-14B as strong models. For each domain, the weak model is trained on 30% of the human-annotated data, and the remaining data is used to align the strong model. The GRA (%) is calculated by the stanfordnlp/SteamSHP-flan-t5-large preference model, comparing aligned models against the SFT baseline. As a result, CW-DPO achieves a 1.4% average GRA improvement compared to the best alternative baselines, WS-DPO, demonstrating its effectiveness across diverse domains and tasks (Table below). We included this result in the revised manuscript.
>
> - ### Evaluation on Diverse Domains
>
> | Domain        | Strong | Human | WS-DPO | CW-DPO |
> |---------------|--------|-------|--------|--------|
> | askacademia   | 7B     | 53.6  | 55.1   | **56.4** |
> |               | 14B    | 50.6  | 53.3   | **55.1** |
> | askbaking     | 7B     | 49.1  | 50.9   | **51.4** |
> |               | 14B    | 56.6  | 54.5   | **58.4** |
> | askengineers  | 7B     | 56.9  | 58.0   | **59.1** |
> |               | 14B    | 54.9  | **57.2** | 55.4 |
> | askphilosophy | 7B     | 58.1  | 59.1   | **60.0** |
> |               | 14B    | 51.0  | **58.2** | 56.8 |
> | **Avg.**      | –      | 53.9  | 55.8   | **56.6** |
>
> > [5] Cu et al. "UltraFeedback: Boosting Language Models with High-quality Feedback." CoRR2023.
> > [6] Kawin et al. "Understanding dataset difficulty with $\mathcal {V} $-usable information." ICML2022.

---

### Official Review · Reviewer_Bgez · 2025-11-03

**Soundness:** 3
**Presentation:** 2
**Contribution:** 3
**Rating:** 6
**Confidence:** 4

**Summary:**

The paper proposes a confidence weighting of preference samples in various preference optimizaiton strategies, inspiried by the fact that even weak LLMs can surpass human annotations when it comes to preference data. The framework first trains a weak LLM to directly output scalar values as the rewards for the prompt and response, which are then used to obtain preference labels on a large amount of unlabeled preference data. During preference optimization, the per-sample loss is weighted based on the confidence of the weak LLM. Empirical validation on three datasets are given.

**Strengths:**

- The idea is useful in practical scenarios and straightforward to apply
- The analysis presented in this paper aligns with them of prior art, whilst also adding further results

**Weaknesses:**

- Most analysis done in this work uses models from the same family. It isn't entirely clear whether the proposed method will generalize across different model families.
- It's unclear on how exactly the weak LLMs are trained. The equation makes it seem like a regression problem, but the footnote says classification. Could you clarify this part?

**Questions:**

See Weaknesses

---

> ### Author Response · Authors · 2025-11-23
>
> We thank the reviewer for thoughtful review, valuable feedback and positive evaluation of our work. We are grateful to the reviewer for recognizing that our method is practical and straightforward to apply, and we are encouraged that the reviewer found our work to provide additional insightful analysis beyond the key findings of the prior baseline.
>
> ---
> **W1.** *Most analysis done in this work uses models from the same family. It isn't entirely clear whether the proposed method will generalize across different model families.*
>
> **Response.** We thank the reviewer for this suggestion. To demonstrate the generalization ability of CW-PO across different model families, we additionally evaluated it in three settings: (1) OPT-125M → Qwen-2.5-14B, (2) Qwen-2.5-0.5B → OPT-13B, and (3) Qwen-2.5-0.5B → Llama-3.1-8B-Instruct. Across Harmlessness, Helpfulness, and HH-RLHF evaluations, CW-DPO outperforms WS-DPO as well as alignment using human-provided annotations (Table below). These results demonstrate that CW-PO generalizes effectively across different model families. We included this result in the revised manuscript.
>
> - ### **Generalization across Different Model Families**
>
> | Dataset     | Human | WS-DPO | CW-DPO |
> |-------------|:-----:|:------:|:------:|
> | **OPT-125M → Qwen-2.5-14B** |  |  |  |
> | Harmless    | 67.3  | 66.4  | **72.2** |
> | Helpful     | 54.1  | 55.9  | **57.2** |
> | HH-RLHF     | 78.8  | 77.7  | **79.7** |
> | **Qwen-2.5-0.5B → OPT-13B** |  |  |  |
> | Harmless    | 74.3  | 72.4  | **75.4** |
> | Helpful     | 65.7  | **75.4** | 73.6 |
> | HH-RLHF     | 56.9  | 57.2  | **62.5** |
> | **Qwen-2.5-0.5B → Llama-3.1-8B-Instruct** |  |  |  |
> | Harmless    | 60.6  | 61.9  | **63.1** |
> | Helpful     | 61.3  | 60.1  | **63.8** |
> | HH-RLHF     | 65.7  | **67.2** | 66.2 |
> | **Avg.**    | 65.0  | 66.0  | **68.2** |
>
> ---
> **W2.** *It's unclear on how exactly the weak LLMs are trained. The equation makes it seem like a regression problem, but the footnote says classification. Could you clarify this part?*
>
> **Response.** We appreciate the reviewer’s question. Although the Bradely-Terry (BT) objective in equation (4) is often written in a regression-like form, it is in fact equivalent to a binary classification loss.
>
> Given a response pair $(y_1, y_2)$, the weak LLM produces scores $\pi_w(x, y_1)$ and $\pi_w(x, y_2)$, and the BT model assigns $y^{+}$ to the more preferable response and $y^{-}$ to the less preferable one. The BT probability is then defined as
>
> $p(y^{+} \succ y^{-} \mid x)= \sigma\left(\pi_w(x, y^{+}) - \pi_w(x, y^{-})\right),$
>
> which has the same mathematical form as the output of a logistic classifier with binary label $t = 1$, indicating that $y^{+}$ is the correct (preferred) response. Consequently, minimizing the BT negative log-likelihood,
>
> $\mathcal{L}_{\mathrm{BT}}= -\log \sigma\left(\pi(x, y^{+}) - \pi(x, y^{-})\right),$
>
> is mathematically equivalent to minimizing the binary cross-entropy loss for a classifier predicting $t = 1$.
>
> To avoid confusion, we revise the description in the script as follows: “Unlike the work of Tao & Li [1], which does not modify the weak LLM’s final layer and computes an implicit reward directly from token probabilities during the response generation process, our approach modifies the weak LLM’s final layer and trains it with BT to predict preferences based on a pairwise logit score. Further details for the calculation of implicit reward are provided in Appendix G.”
>
> > [1] Tao & Li. "Your Weak LLM is Secretly a Strong Teacher for Alignment." ICLR2025.

---

### Comment · Area_Chair_KEmy · 2025-11-26

Dear Reviewers,

Thank you for sharing your valuable insights and expertise, which have played an important role in the review process.

In response to the initial feedback, the authors have submitted a detailed rebuttal addressing the comments raised by the reviewers.

I would appreciate it if you could carefully review their response and consider how it may affect your initial evaluation.

Please feel free to share your updated thoughts or any additional comments after reviewing the rebuttal.

Thank you again for your time and contributions.

---

### Author Response · Authors · 2025-12-03

We thank all reviewers for their thoughtful reviews and valuable feedback that helped to strengthen our work. We are grateful for their recognition that CW-PO is **novel** (zg8q, wGU3), **useful in practical scenarios**, and straightforward to apply (Bgez, zg8q, wGU3), while also **substantially reducing computational cost** (zg8q, wGU3). We appreciate the reviewers’ acknowledgement of our contribution to provide **insightful analysis** and that CW-PO aligns well with this analysis (Bgez, vgGH, zg8q). Additionally, we value the recognition of the **strong performance** demonstrated through extensive evaluation (z1Ex, zg8q), as well as the method’s **practicality as a plug-and-play enhancement** to existing preference optimization approaches (wGU3).

During the rebuttal, we conducted additional **empirical evaluations** in response to the reviewers' feedback.
- The common comment of three reviewers was to demonstrate generalizability of the CW-PO across different model families. To address this, we added experiments using three different model families, OPT, Qwen, and Llama and showed the effectiveness of the CW-PO framework using different models.

- A common comment from two reviewers was the robustness of the CW-PO under noisy and imbalanced tasks. In response, we included additional experiments where we examine the impact of adversarially biased (poisoned) human data and the impact of imbalanced weak-model training data. We show that the CW-DPO is more robust to such imbalance and under noisy labelings compared to the baseline WS-DPO.

- Two reviewers asked for the qualitative analysis where the weak LLM is highly confident but human annotators might disagree or be inconsistent. We provided examples of different confidence scores from weak LLM and showed that high-confidence samples are clearly distinguishable compared to low-confidence samples.

- The concern from reviewer z1Ex was the lack of comparison to the PairRM framework. We included this baseline during the rebuttal and showed that CW-PO outperforms this baseline across different datasets by a large margin.

- In addition, we tested the performance in the online training setting, expanded evaluation on additional preference-alignment domains (SHP) and performed analysis of variance and sampling robustness.

Finally, we addressed all clarification questions and expanded our related work with the papers suggested by the reviewers and detailed explanations of the difference between the suggested papers and our work.

We thank the reviewers and area chairs for their efforts and valuable feedback that helped to further strengthen our manuscript. We hope that our detailed rebuttal, additional experiments and  the revised address the reviewers’ concerns and demonstrate that the CW-PO is a highly effective and efficient framework for preference alignment which requires only a small amount of human annotations.

---

### Meta-Review · Area_Chair_JppH · 2025-12-14

**Summary:**

The reviews are overall positive. There are two major concerns from two reviewers. First, how the proposed method generalizes for other online RL frameworks. Second, whether it can handle other datasets/tasks. With the additional details and experiments provided by the authors, I think these two concerns have been addressed reasonably. Thus, I recommend accept.

**Reviewer Concerns:**

Most concerns have been addressed with additional details and experimental results provided.

**Reviewer Scores:**

Better if Reviewer z1Ex and Reviewer wGU3 can read the rebuttal and re-consider their scores

---

### Decision · Program_Chairs · 2026-01-26

Accept (Poster)